# Accelerating Goal-Conditioned Reinforcement Learning Algorithms and Research

**Michał Bortkiewicz**[1]   **Władysław Pałucki**[2]   **Vivek Myers**[3]
**Tadeusz Dziarmaga**[4]   **Tomasz Arczewski**[4]
**Łukasz Kuciński**[2,5,6]   **Benjamin Eysenbach**[7]

[1]Warsaw University of Technology   [2]University of Warsaw   [3]UC Berkeley
[4]Jagiellonian University   [5]Polish Academy of Sciences
[6]IDEAS NCBR   [7]Princeton University

 michalbortkiewicz8@gmail.com   wladek.palucki@gmail.com

## Abstract

Self-supervision has the potential to transform reinforcement learning (RL), paralleling the breakthroughs it has enabled in other areas of machine learning. While self-supervised learning in other domains aims to find patterns in a fixed dataset, self-supervised goal-conditioneds reinforcement learning (GCRL) agents discover *new* behaviors by learning from the goals achieved during unstructured interaction with the environment. However, these methods have failed to see similar success, both due to a lack of data from slow environment simulations as well as a lack of stable algorithms. We take a step toward addressing both of these issues by releasing a high-performance codebase and benchmark (`JaxGCRL`) for self-supervised GCRL, enabling researchers to train agents for millions of environment steps in minutes on a single GPU. By utilizing GPU-accelerated replay buffers, environments, and a stable contrastive RL algorithm, we reduce training time by up to $22\times$. Additionally, we assess key design choices in contrastive RL, identifying those that most effectively stabilize and enhance training performance. With this approach, we provide a foundation for future research in self-supervised GCRL, enabling researchers to quickly iterate on new ideas and evaluate them in diverse and challenging environments. Code: https://github.com/MichalBortkiewicz/JaxGCRL.

## 1 Introduction

Self-supervised learning has significantly influenced machine learning over the last decade, transforming how research is done in domains such as natural language processing and computer vision (Chen et al., 2020b; Dosovitskiy et al., 2020; Vaswani et al., 2017). In the context of reinforcement learning (RL), most self-supervised prior methods apply the same recipe that has been successful in other domains: learning representations or models from a large, fixed dataset (Hoffmann et al., 2022; Sardana et al., 2024; Muennighoff et al., 2023). However, the RL setting also enables a fundamentally different type of self-supervised learning: rather than learning from a fixed dataset (as done in NLP and computer vision), a self-supervised *reinforcement* learner can collect its own dataset. Thus, rather than learning a representation of a dataset, the self-supervised reinforcement learner acquires a representation of an environment or of behaviors and optimal policies therein. Self-supervised reinforcement learners may address many of the challenges that stymie today's foundation's models: reasoning about the consequences of actions (Rajani et al., 2019; Kwon et al., 2023) (i.e., counterfactuals (Bhargava and Ng, 2022; Jin et al., 2023)) and long horizon planning (Bhargava and Ng, 2022; Du et al., 2023; Guan et al., 2023).

In this paper, we study self-supervised RL in an online setting: an agent interacts in an environment without a reward to learn representations, which are later used to quickly solve downstream tasks. We focus on goal-conditioned reinforcement learning (GCRL) algorithms, which aim to use these unsupervised interactions to learn policies for achieving various goals – an essential capability for

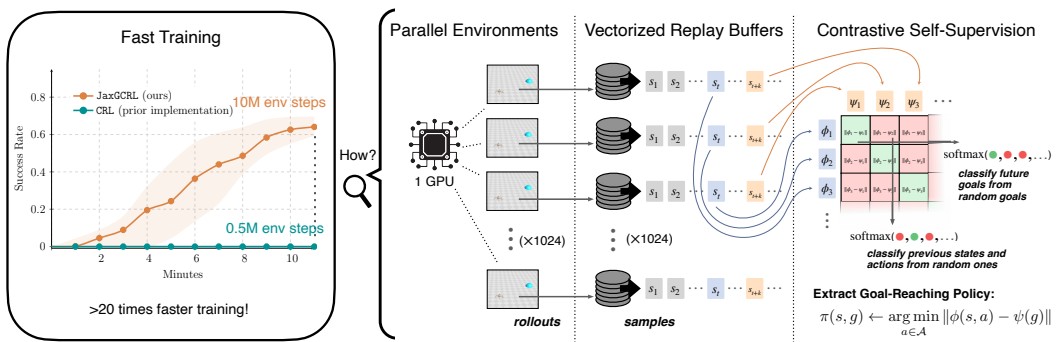

Figure 1: `JaxGCRL` **is fast.** It learns goal-reaching policies for Ant in 10 minutes on 1 GPU. This paper releases a GCRL benchmark and baseline algorithms that enable research and experiments to be done in minutes.

multipurpose robots. Prior work has proposed several algorithms for self-supervised RL (Eysenbach et al., 2022; Zheng et al., 2024; Myers et al., 2023; 2024a), including algorithms that focus on learning goals (Erraqabi et al., 2022). However, these methods were limited by small datasets and infrequent online interactions with the environment, which has prevented them from exploring the potential emergent properties of self-supervised reinforcement learning on large-scale data.

The main goal of this paper is to introduce `JaxGCRL`: an extremely *fast* GPU-accelerated codebase and benchmark for effective self-supervised GCRL research. For instance, an experiment with 10 million environment steps lasts only around 10 minutes on a single GPU, which is **up to 22× faster** than in the original contrastive RL codebase (Eysenbach et al., 2022) (Fig. 1). This speed allows researchers to "interactively" debug their algorithms, changing pieces and getting results in near real-time for multiple random seeds on a single GPU without the hustle of distributed training. Consequently, `JaxGCRL` eliminates the barriers to entry to state-of-the-art GCRL research, making it more accessible to under-resourced institutions.

To achieve this training and performance improvement, we combine insights from self-supervised RL with recent advances in GPU-accelerated simulation. The *first key ingredient* is recent work on GPU-accelerated simulators, both for physics (Freeman et al., 2021; Thibault et al., 2024; Liang et al., 2018) and other tasks (Matthews et al., 2024; Dalton et al., 2020; Fischer et al., 2009; Bonnet et al., 2024; Rutherford et al., 2024) that enable users to collect data up to 1000× faster (Freeman et al., 2021) than prior methods based on CPUs. The *second key ingredient* is a highly stable algorithm build upon recent work on contrastive RL (CRL) (Eysenbach et al., 2022; Zheng et al., 2024), which uses temporal contrastive learning to learn a value function. The *third key ingredient* is a suite of tasks for evaluating self-supervised RL agents which is not only blazing fast but also stress tests the exploration and long-horizon reasoning capabilities of RL policies.

The contributions of this work are as follows:

**`JaxGCRL` codebase:** a blazingly fast JIT-compiled training pipeline for GCRL experiments.

**`JaxGCRL` benchmark:** we introduce a suite of 8 GPU-accelerated state-based environments that help to accurately assess GCRL algorithm capabilities and limitations.

**Extensive empirical analysis:** we evaluate important CRL design choices, focusing on key algorithm components, architecture scaling, and training in data-rich settings.

## 2 RELATED WORK

We build upon recent advances in GCRL, self-supervised RL, and hardware-accelerated physics simulators, showing that CRL enables fast and reliable training across a diverse suite of environments.

### 2.1 GOAL-CONDITIONED REINFORCEMENT LEARNING

GCRL is a special case of the general multi-task RL setting, in which the potential tasks are defined by goal states that the agent is trying to reach in an environment (Kaelbling, 1993; Ghosh et al., 2019;

Nair et al., 2018). Achieving any goal is appealing for generalist agents, as it allows for diverse behaviors without needing specific reward functions for each task (each state defines a task when seen as a goal) (Schaul et al., 2015). GCRL techniques have seen success in domains such as robotic manipulation (Andrychowicz et al., 2017; Eysenbach et al., 2021; Ghosh et al., 2021; Ding et al., 2022; Walke et al., 2023) and navigation (Shah et al., 2023; Levine and Shah, 2023; Manderson et al., 2020; Hoang et al., 2021). Recent work has shown that representations of goals can be imbued with additional structure to enable capabilities such as language grounding (Myers et al., 2023; Ma et al., 2023), compositionality (Liu et al., 2023; Myers et al., 2024b; Wang et al., 2023), and planning (Park et al., 2023; Eysenbach et al., 2024). We show how GCRL techniques based on contrastive learning (Eysenbach et al., 2022) can be scaled with GPU acceleration to enable fast and stable training.

## 2.2 ACCELERATING DEEP REINFORCEMENT LEARNING

Deep RL has only recently become practical for many tasks, in part due to improvements in hardware support for these algorithms. Distributed training has enabled RL algorithms to scale across hundreds of GPUs (Mnih et al., 2016; Espeholt et al., 2018; 2020; Hoffman et al., 2022). To resolve the bottleneck of environment interaction with CPU-bound environments, various GPU-accelerated environments have been proposed (Matthews et al., 2024; Freeman et al., 2021; Bonnet et al., 2024; Rutherford et al., 2024; Liang et al., 2018; Dalton et al., 2020; Makoviychuk et al., 2021; Lange, 2022; Lu et al., 2022). Most of these works rely on JAX (Bradbury et al., 2018; Heek et al., 2023; Hennigan et al., 2020), which enables JIT compilation, operator fusion and other components necessary for efficient vectorized code execution. These features significantly accelerate data collection by supporting rollouts in hundreds of parallelized environments. We build on these advances to scale self-supervised RL to data-rich settings.

## 2.3 SELF-SUPERVISED RL

Self-supervised training has enabled key breakthroughs in language modeling and computer vision (Sermanet et al., 2017; Zhu et al., 2020; Devlin et al., 2019; He et al., 2022; Mikolov et al., 2013). In the context of RL, by the term "self-supervised", we mean techniques that can be learned through interaction with an environment without a reward signal. Perhaps the most successful form of self-supervision has been in multi-agent games that can be rapidly simulated, such as Go and Chess, where self-play has enabled the creation of superhuman agents (Silver et al., 2016; 2017; Zha et al., 2021). When learning goal-reaching agents, another basic form of self-supervision is to relabel trajectories as successful demonstrations of the goal that was reached, even if it differs from the original commanded goal (Kaelbling, 1993; Venkattaramanujam et al., 2020). This technique has seen recent adoption as "hindsight experience replay" for various deep RL algorithms (Andrychowicz et al., 2017; Ghosh et al., 2021; Chebotar et al., 2021; Rauber et al., 2021; Eysenbach et al., 2022).

Another perspective on self-supervision is intrinsic motivation, broadly defined as when an agent computes its own reward signal (Barto, 2013). Intrinsic motivation methods include curiosity (Barto, 2013; Bellemare et al., 2016; Baumli et al., 2021), surprise minimization (Berseth et al., 2019; Rhinehart et al., 2021), and empowerment (Klyubin et al., 2005; de Abril and Kanai, 2018; Choi et al., 2021; Myers et al., 2024a). Closely related are skill discovery methods, which aim to construct intrinsic rewards for diverse collections of behavior (Gregor et al., 2016; Eysenbach et al., 2019; Sharma et al., 2020; Kim et al., 2021; Park et al., 2021; Laskin et al., 2022; Park et al., 2024). Self-supervised RL methods have been difficult to scale due to the need for many environment interactions (Franke et al., 2021; Mnih et al., 2015). This work addresses that challenge by offering a fast and scalable contrastive RL algorithm on a benchmark of diverse tasks.

## 2.4 RL BENCHMARKS

The RL community has recently started to pay greater attention to how RL research is conducted, reported, and evaluated (Henderson et al., 2018; Jordan et al., 2020; Agarwal et al., 2022). A key issue is a lack of reliable and efficient benchmarks: it becomes hard to rigorously compare novel methods when the number of trials needed to see statistically significant results across diverse settings ranges in the thousands of training hours (Jordan et al., 2024). Some benchmarks that *have* seen adoption include OpenAI gym/Gymnasium (Brockman et al., 2022; Towers et al., 2024), DeepMind Control Suite (Tassa et al., 2018), and D4RL (Fu et al., 2021). More recently, hardware-accelerated

versions of some of these benchmarks have been proposed (Gu et al., 2021; Matthews et al., 2024; Koyamada et al., 2023; Makoviychuk et al., 2021; Nikulin et al., 2023). However, the RL community still lacks advanced benchmarks for goal-conditioned methods. We address this gap with JaxGCRL, significantly lowering the GCRL evaluation cost and thereby enabling impactful RL research.

## 3 PRELIMINARIES

In this section, we introduce notation and preliminary definitions for goal-conditioned RL and the contrastive RL method, which serves as the foundation for this work.

In the goal-conditioned reinforcement learning setting, an agent interacts with a controlled Markov process (CMP) $\mathcal{M} = (\mathcal{S}, \mathcal{A}, p, p_0, \gamma)$ to reach arbitrary goals (Kaelbling, 1993; Andrychowicz et al., 2017; Blier et al., 2021). At any time $t$ the agent will observe a state $s_t$ and select a corresponding action $a_t \in \mathcal{A}$. The dynamics of this interaction are defined by the distribution $p(s_{t+1} \mid s_t, a_t)$, with an initial distribution $p_0(s_0)$ over the state at the start of a trajectory, for $s_t \in \mathcal{S}$ and $a_t \in \mathcal{A}$.

For any goal $g \in \mathcal{S}$, we cast optimal goal-reaching as a problem of inference (Borsa et al., 2019; Barreto et al., 2022; Blier et al., 2021; Eysenbach et al., 2022): given the current state and desired goal, what is the most likely action that will bring us toward that goal? As we will see, this is equivalent to solving the Markov decision process (MDP) $\mathcal{M}_g$ obtained by augmenting $\mathcal{M}$ with the goal-conditioned reward $r_g(s_t, a_t) \triangleq (1 - \gamma)\gamma p(s_{t+1} = g \mid s_t, a_t)$. Formally, a goal-conditioned policy $\pi(a \mid s, g)$ receives both the current observation of the environment as well as a goal $g \in \mathcal{S}$.

We denote the $k$-step action-conditioned policy distribution $p_k^\pi(s_k \mid s_0, a_0)$ as the distribution of states $k$ steps in the future given the initial state $s_0$ and action $a_0$ under $\pi$. We define the discounted state visitation distribution as $p_\gamma^\pi(s^+ \mid s, a) \triangleq (1 - \gamma)\sum_{t=0}^{\infty} \gamma^t p_t^\pi(s^+ \mid s, a)$, which we interpret as the distribution of the state $T$ steps in the future for $T \sim \text{Geom}(1 - \gamma)$. This last expression is precisely the $Q$-function of the policy $\pi(\cdot \mid \cdot, g)$ for the reward $r_g$: $Q_g^\pi(s, a) \triangleq p_\gamma^\pi(g \mid s, a)$ (see Appendix D.1). For a given distribution over goals $g \sim p_\mathcal{G}$, we can now write the overall objective as

$$\max_{\pi(\cdot \mid \cdot, \cdot)} \mathbb{E}_{p_0(s_0) p_\mathcal{G}(g) \pi(a_0 \mid s_0, g)} \left[ p_\gamma^\pi(g \mid s_0, a_0) \right]. \tag{1}$$

### 3.1 CONTRASTIVE CRITIC LEARNING

CRL is an actor-critic method that aims to solve Eq. (1). The critic is represented as a state-action-goal value function $f(s, a, g)$, which provides the likelihood of future states and how various actions influence the likelihood of future states. This function satisfies:

$$f(s, a, g) \propto p_\gamma^\pi(g \mid s, a) = Q_g^\pi(s, a). \tag{2}$$

Therefore, we can treat $f(s, a, g)$ as an approximation of the Q-function and use it to train the actor. This approach builds on previous research that frames learning of this critic as a classification problem (Eysenbach et al., 2022; 2021; Zheng et al., 2023; 2024; Farebrother et al., 2024; Myers et al., 2024b). Training is performed on batches of $(s, a, g)$ to classify whether or not $g$ is the future state corresponding to the trajectory starting from $(s, a)$. Thus, in each sample from batch $(s_i, a_i, g_i) \in \mathcal{B}$ the goal $g_i$ is sampled from the future of the trajectory containing $(s_i, a_i)$.

The family of CRL algorithms consists of the following components: (a) the state-action pair and goal state representations, $\phi(s, a)$ and $\psi(g)$, respectively; (b) the critic, which is defined as an energy function $f_{\phi,\psi}(s, a, g)$, measuring *some* form of similarity between $\phi(s, a)$ and $\psi(g)$; and (c) a contrastive loss function, which is a function of the matrix containing the critic values $\{f_{\phi,\psi}(s_i, a_i, g_j)_{i,j}\}$ over the elements of the batch $\mathcal{B}$. The base contrastive loss we study will be the infoNCE objective (Sohn, 2016), modified to use a symmetrized (Radford et al., 2021) critic parameterized with $\ell_2$-distances (Eysenbach et al., 2024). The final objective for the critic can thus be expressed as:

$$\min_{\phi,\psi} \mathbb{E}_\mathcal{B} \left[ -\sum_{i=1}^{|\mathcal{B}|} \log\left( \frac{e^{f_{\phi,\psi}(s_i, a_i, g_i)}}{\sum_{j=1}^{K} e^{f_{\phi,\psi}(s_i, a_i, g_j)}} \right) - \sum_{i=1}^{|\mathcal{B}|} \log\left( \frac{e^{f_{\phi,\psi}(s_i, a_i, g_i)}}{\sum_{j=1}^{K} e^{f_{\phi,\psi}(s_j, a_j, g_i)}} \right) \right],$$

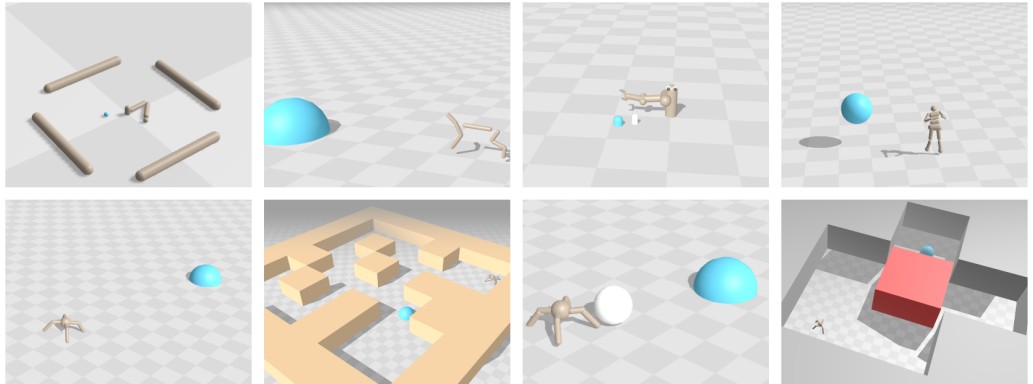

Figure 2: `JaxGCRL` benchmark: New suite of GPU-accelerated environments for studying GCRL. In this setting, the agent does not receive any rewards or demonstrations, making some of these tasks an excellent testbed for studying exploration and long-horizon reasoning. Our accompanying implementation of GCRL algorithms trains with more than 15K environment steps per second on a single GPU, enabling rapid experimentation.

where

$$f_{\phi,\psi}(s, a, g) = \|\phi(s, a) - \psi(g)\|_2.$$

This loss contrasts each positive sample with the batch of negative samples. Other losses, which are tested in Section 5.3, are further discussed in Appendix A.2.

## 3.2 POLICY LEARNING

We use a DDPG-style policy extraction loss to learn a goal-conditioned policy, $\pi_\theta(a|s, g)$, by optimizing the critic $f_{\phi,\psi}$ (Lillicrap et al., 2016):

$$\max_\theta \mathbb{E}_{p(s,a)p(g|s,a)\pi_\theta(a'|s,g)} \left[ f_{\phi,\psi}(s, a', g) \right] \qquad (3)$$

Each batch is formed by sampling $(s, a)$ pairs uniformly and then sampling goals $g$ from the states that occur after $s$ in a trajectory.

## 4 JAXGCRL: A NEW BENCHMARK AND IMPLEMENTATION

JaxGCRL is an efficient tool for developing and evaluating new GCRL algorithms. It shifts the bottleneck from compute to implementation time, allowing researchers to test new ideas, like contrastive objectives, within minutes—a significant improvement over traditional RL workflows.

### 4.1 JAXGCRL SPEEDUP ON A SINGLE GPU

We compare the proposed fully JIT-compiled implementation of CRL in `JaxGCRL` to the original implementation from Eysenbach et al. (2022). In particular, Fig. 1 shows the performance in `Ant` environment along with the experiment's wall-clock time. The speedup in this configuration is **22-fold**, with the new implementation reaching a training speed of over 16500 environment steps per second with a 1:16 update to data (UTD) ratio. For results with other UTD, refer to Section 5.6. Complete speedup results are provided in Appendix A.1. Importantly, BRAX physics simulator differs from the original MuJoCo, so performance numbers here may vary slightly from prior work.

### 4.2 JAXGCRL ENVIRONMENTS IN THE BENCHMARK

To evaluate the performance of GCRL methods, we propose `JaxGCRL` benchmark consisting of 8 diverse continuous control environments. These environments range from simple, ideal for quick checks, to complex, requiring long-term planning and exploration for success. The following list provides a brief description of each environment, with the technical details summarized in Table 1:

**Reacher (Brockman et al., 2022).** A 2D manipulation task involves positioning the end part of a 2-segment robotic arm in a specific location sampled uniformly from a disk around an agent.

**Half Cheetah (Wawrzyński, 2009).** In this 2d task, a 2-legged agent has to get to a goal that is sampled from one of the 2 possible places, one on each side of the starting position.

**Pusher (Brockman et al., 2022).** This is a 3d robotic task, that consists of a robotic arm and a movable circular object resting on the ground. The goal is to use the arm to push the object into the goal position. With each reset, both position of the goal and movable object are selected randomly.

**Ant (Schulman et al., 2016).** A re-implementation of the MuJoCo Ant with a quadruped robot that needs to walk to the goal randomly sampled from a circle centred at the starting position.

**Ant Maze (Fu et al., 2021).** This environment uses the same quadruped model as the previous Ant, but the agent must navigate a maze to reach the target. We prepared 3 different mazes varying in size and difficulty. In each maze, the goals are sampled from a set of listed possible positions.

**Ant Soccer (Tunyasuvunakool et al., 2020).** In this environment, the Ant has to push a spherical object into a goal position that is sampled uniformly from a circle around the starting position. The position of the movable sphere is randomized on the line between an agent and a goal.

**Ant Push (Fu et al., 2021).** To reach a goal, the Ant has to push a movable box out of the way. If the box is pushed in the wrong direction, the task becomes unsolvable. Succeeding requires exploration, understanding block dynamics, and how it changes the layout of the maze.

**Humanoid (Tassa et al., 2012).** Re-implementing the Mujoco task involves navigating a complex humanoid-like robot to walk towards a goal sampled from a disk centred at the starting position.

In Fig. 3, we report the baseline results for the proposed benchmark. It is worth noting that for most environments, experiments involving 50M environment steps can be completed in less than an hour.

## 4.3 CONTRASTIVE RL DESIGN CHOICES

`JaxGCRL` streamlines and accelerates the evaluation of new GCRL algorithms, enabling quick assessment of key CRL design choices:

**Energy functions.** Measuring the similarity between samples can be achieved in various ways, resulting in potentially different agent behaviors. Our analysis in following sections include cosine similarity (Chen et al., 2020a), dot product (Radford et al., 2021), and negative $L_1$ and $L_2$ distance (Hu et al., 2023), detailed list can be found in Appendix A.2. Even though there is no consensus on the choice of energy functions for temporal representations, recent works showed that they should abide by quasimetric properties (Wang et al., 2023; Myers et al., 2024b).

**Contrastive losses.** Beside InfoNCE-type losses (van den Oord et al., 2019; Eysenbach et al., 2022), we evaluate FlatNCE-like losses (Chen et al., 2021), and a Monte Carlo version of Forward-Backward unsupervised loss (Touati and Ollivier, 2021). Additionally, we test novel objectives inspired by preference optimization for large language models (Calandriello et al., 2024). Specifically, we evaluate DPO, IPO, and SPPO, which increase the scores of positive samples and reduce the scores of negative ones. A full list of contrastive objectives can be found in Appendix A.2.

**Architecture scaling.** Scaling neural network architectures to improve performance is a common practice in other areas of deep learning, but it remains relatively underexplored in RL models, as noted in Nauman et al. (2024a;b). Recently, Zheng et al. (2024) showed that CRL might benefit from deeper and wider architectures with Layer Normalization (Ba et al., 2016) for offline CRL in pixel-based environments; we want to examine whether this also holds for online state-based settings.

## 5 EXAMPLES OF FAST EXPERIMENTS POSSIBLE WITH THE NEW BENCHMARK

The goal of our experiments is twofold: (1) to establish a baseline for the proposed `JaxGCRL` environments, and (2) to evaluate CRL performance in relation to key design choices. In Section 5.1, we define setup that is used for most of the experiments unless explicitly stated otherwise. *First* in Section 5.2, we report baseline results on `JaxGCRL`. *Second*, in Sections 5.3 and 5.4, we try to understand the influence of design choices on CRL learning performance. *Third*, in Section 5.5, we asses those design choices in a data-rich setting with 300M environment steps. *Lastly*, in Section 5.6, we explore the relation between performance and UTD.

### 5.1 EXPERIMENTAL SETUP

Our experiments use `JaxGCRL` suite of simulated environments described in Section 4.2. We evaluate algorithms in an online setting for 50M environment steps. We compare CRL with Soft Actor-Critic (SAC) (Haarnoja et al., 2018), SAC with Hindsight Experience Replay (HER) (Andrychowicz et al., 2017), TD3 (Fujimoto et al., 2018), TD3+HER, and PPO (Schulman et al., 2017). For algorithms with HER, we use *final* strategy relabelling, i.e. relabeling goals with states achieved at the end of the trajectory. In the majority of experiments, we use CRL with L2 energy function, symmetric InfoNCE objective, and a tuneable entropy coefficient for all methods. See Appendix B for details. We use two performance metrics: success rate and time near goal. Success rate measures whether the agent reached the goal at least once during the episode, while time near goal indicates how long the agent stayed close to it. We use a sparse reward for all baselines, with $r = 1$ when the agent is in goal proximity and $r = 0$ otherwise. We define goal-reaching as achieving proximity below the `goal distance` threshold defined in Table 1. The implementations of PPO and SAC are sourced from the Brax repository (Freeman et al., 2021), while TD3, HER, and CRL are partially based on Brax.

### 5.2 `JaxGCRL` BENCHMARK RESULTS

We establish baseline results for `JaxGCRL` with all algorithms in Fig. 3. Clearly, CRL achieves the highest performance across tested methods, resulting in non-trivial policies even in the hardest tasks in terms of high-dimensional state and action spaces (Humanoid) and exploration (Pusher and Ant Push). However, the performance in these challenging environments is low, indicating room for improvement for future contrastive RL methods. As expected, HER can improve performance for both TD3 and SAC. In contrast, PPO performs poorly across all tasks, likely due to the challenges posed by the sparse reward setting. Additional experiments on `JaxGCRL` environments with design choices discussed in the following sections can be found in Appendix A.4.

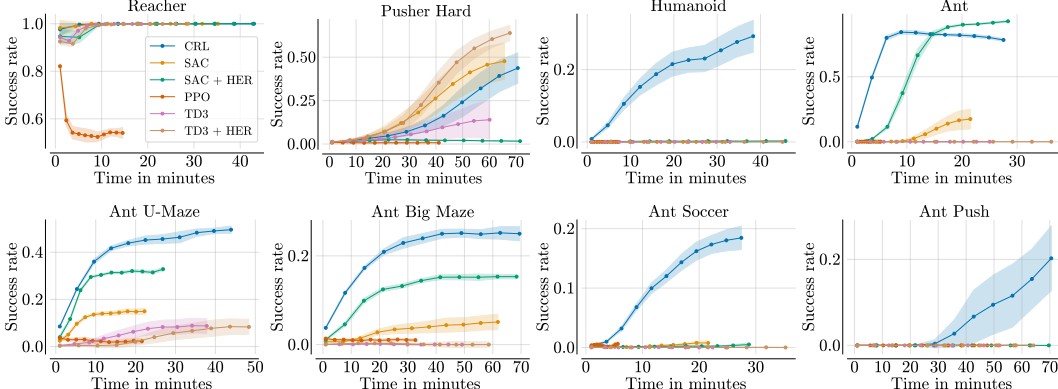

Figure 3: **Baseline results in JaxGCRL benchmark.** Success rates of all the baseline algorithms for 50M environment steps for every `JaxGCRL` environment. CRL outperforms other baselines in most of the environments. The training speed is a function of the environment complexity, method complexity, and physics backend; see Appendix A.4. Specifically, due to differences in how each method works, the speed varies greatly in the same environments; this can be best seen with the PPO method being significantly faster than others due to it not using a replay buffer, which frees up GPU memory for more parallel environment simulations. Results are reported as the interquartile mean (IQM) along with its standard error, based on 10 seeds.

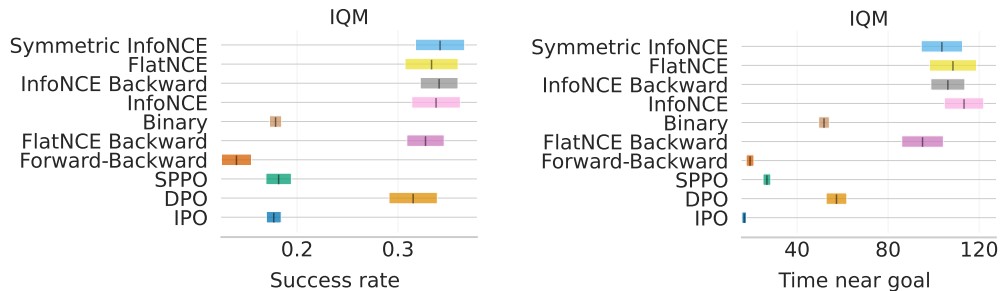

Figure 4: **InfoNCE-based loss functions perform best.** The critic loss functions that achieve the highest success rates are based on InfoNCE and DPO. However, DPO policies tend to stay at the goal for a shorter duration. IQMs averaged over 10 seeds and plotted with one standard error.

## 5.3 Contrastive objectives and energy functions comparison

Contrastive objective and energy function are the two main components of contrastive methods, serving as the primary drivers of their final performance. We evaluate CRL with 10 different contrastive objectives, as defined in Section 4.3 across five environments: Ant Soccer, Ant, Ant Big Maze, Ant U-Maze, Ant Push, and Pusher, and report aggregated performance. For the energy function, we use L2, as it consistently resulted in the highest performance for the CRL method, especially regarding time near goal, see Appendix A.2. Additionally, we apply logsumexp regularization with a coefficient of 0.1 to each objective. This auxiliary objective is essential, as without it, the performance of InfoNCE deteriorates significantly (Eysenbach et al., 2022). The analysis presented in Fig. 4 reveals that the originally proposed NCE-binary objective, along with forward-backward, IPO, and SPPO, are the least effective objectives among those evaluated. However, for other InfoNCE-derived objectives, it is difficult to determine the best one, as their performance is similar. Interestingly, CRL seems fairly robust to the choice of contrastive objective.

## 5.4 Scaling the Architecture

In this section, we explore how increasing the size of actor and critic networks, in terms of both depth and width, influences CRL performance. We evaluate the aggregated performance during the final 10M steps of a 50 million-step training process across three environments: Ant, Ant Soccer, and Ant U-Maze. We use the L2, Symmetric InfoNCE and logsumexp regularizer coefficient of 0.1 for all architecture sizes.

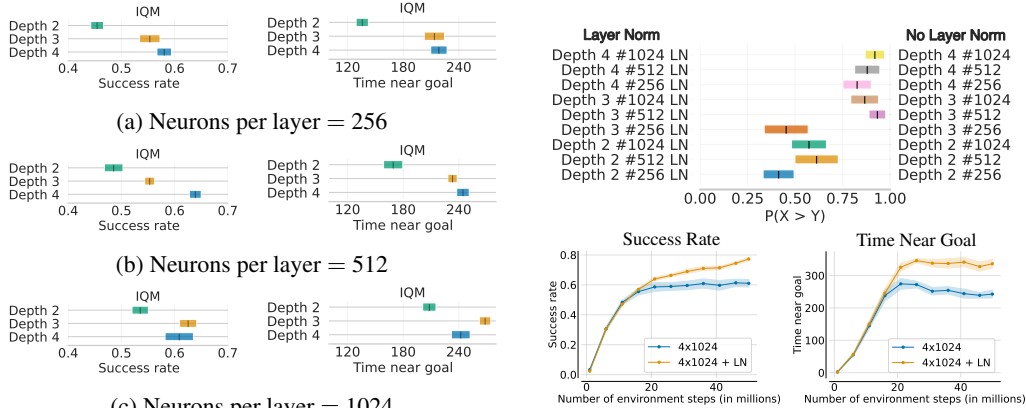

Figure 5: **Scaling the critic and actor networks.** Increasing the width and depth generally enhances performance, but performance levels off for deeper architectures at a width of 1024. Aggregated metrics, 5 seeds per configuration.

Figure 6: **Layer normalization enables stable performance improvement.** Using Layer Normalization (LN) in the largest architecture allows for continued learning even after reaching the saturation point of a standard large architecture.

We present the results of this scaling experiment in Fig. 5 and observe that increasing both the width and depth tends to increase performance. However, performance does not further improve when increasing the depth for width = 1024 neurons. Our next experiment studies whether layer normalization can stabilize the performance of these biggest networks (width of 1024 neurons, depth of 4). Indeed, the results in Fig. 6 show that adding layer normalization before every activation allows better scaling properties, especially for bigger networks.

## 5.5 SCALING THE DATA

We evaluate the benefits CRL gains from training in a data-rich setting. In particular, we report performance for large architectures (studied in Section 5.4) with different combinations of energy functions and contrastive objectives for 300M environment steps in Fig. 7. We observe that the L2 energy function with InfoNCE objective configuration outperforms all others by a substantial margin, leading to a higher success rate and time near the goal across three locomotion tasks. Interestingly, the dot product energy function performs best in the object manipulation task (Ant Soccer). This indicates that only a subset of a wide array of design choices performs well when scaling CRL. Additionally, there is still room for improvement in scaling CRL with data, as the success rate in Ant Soccer and Ant Big Maze remains around 40%. For additional experiments, refer to Appendix A.6

## 5.6 GRADIENT UPDATES TO DATA RATIO

`JaxGCRL` enables efficient execution of extensive experiments. Leveraging this capability, we explore the effect of the model's update frequency in CRL by evaluating a range of UTD ratios. In particular, we examine ratios (1:1, 1:8, 1:16, 1:24, 1:32, 1:49, 1:48) in five environments: Ant, Ant Soccer, Ant U-Maze, Pusher Hard, and Ant Push. Interestingly, we only observe a significant increase in performance for Pusher Hard with a higher number of updates, while in other environments, it leads to decreased or similar performance. With a UTD ratio of 1:16, our code is 22× faster than prior implementations, and with a lower frequency of gradient updates, it can be even faster.

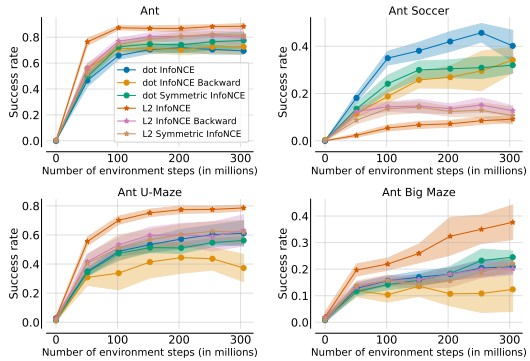

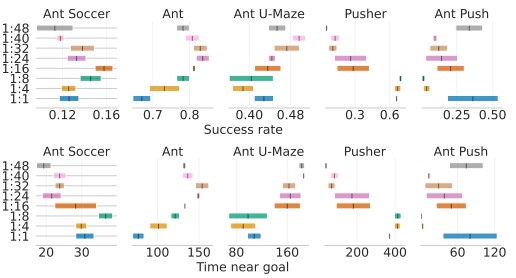

Figure 7: `JaxGCRL` **allows researchers to study energy functions and critic losses over hundreds of millions of steps.** Among tested configurations, L2 with InfoNCE objective performs best in locomotion environments when data is abundant.

Figure 8: **More gradients ≠ better performance.** Success rate (top) and Time near goal (bottom) for different UTD ratios. Increasing the UTD ratio for CRL increases performance only for Pusher.

**Key takeaways from empirical experiments:**

- **Experiments with 10M steps can be completed in minutes, while those with billions of environment steps can be done in a few hours using `JaxGCRL` on a single GPU.**

- CRL is the only method that can learn effectively in all proposed environments without needing a high UTD ratio in most cases. It benefits greatly from using large architectures, especially when Layer Normalization is applied.

- Different combinations of energy and contrastive functions lead to different outcomes: some primarily improve success rates, while others extend the time spent near the goal.

Taken together, these experiments not only provide guidance on good design decisions for self-supervised RL, but also highlight how our fast codebase and benchmark can enable researchers to quickly iterate on ideas and hyperparameters.

## 6    CONCLUSION

In this paper, we introduce `JaxGCRL`, a very fast benchmark and codebase for goal-conditioned RL. The speed of the new benchmark enables us to rapidly study design choices for state-based CRL, including the network architectures and contrastive losses. We expect that self-supervised RL methods will open the door to entirely new learning algorithms with broad capabilities that go beyond the capabilities of today's foundational models. The key step towards this goal is accelerating and democratising self-supervised RL research so that any lab can carry it out regardless of its computing capabilities. Open-sourcing the proposed codebase with easy-to-implement self-supervised RL methods is an important step in this direction.

**Limitations.**    The GCRL paradigm complicates the process of defining goals that are not easily expressed as a single state, making it infeasible for some applications. Additionally, our benchmark environments and methods assume full observability and that goals are being sampled from known goal distribution during training rollouts. Future work should relax these assumptions to make self-supervised RL agents useful in more practical settings. We also only investigate online GCRL settings.

**Reproducibility Statement.**    All experiments can be replicated using the provided publicly available `JaxGCRL` code at https://github.com/MichalBortkiewicz/JaxGCRL. This repository includes comprehensive instructions for setting up the environment, running the experiments, and evaluating the results, making it straightforward to reproduce the findings.

**Acknowledgments.**    This research was substantially supported by the National Science Centre, Poland (grant no. 2023/51/D/ST6/01609), and the Warsaw University of Technology through the Excellence Initiative: Research University (IDUB) program. We gratefully acknowledge the Polish high-performance computing infrastructure, PCSS PLCloud, for providing computational resources and support under grant no. pl0334-01, and PLGrid (HPC Center: ACK Cyfronet AGH), for providing resources and support under grant no. PLG/2024/017040. We would also like to recognize the founding of the DoD NDSEG fellowship for Vivek Myers. This work was partially conducted using Princeton Research Computing resources at Princeton University, a consortium led by the Princeton Institute for Computational Science and Engineering (PICSciE) and Research Computing.

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

# A  ADDITIONAL RESULTS

## A.1  SPEEDUP COMPARISON ACROSS VARIOUS NUMBERS OF PARALLEL ENVIRONMENTS

We present an extended version of the plot from Fig. 1, with additional experiments, as depicted on the left side of Fig. 9. Each experiment was run for 10M environment steps, and for the original repository, we varied the number of parallel actors for data collection, testing configurations with 4, 8, 16, and 32 actors, each running on separate CPU threads. Each configuration was tested with three different random seeds, and we present the results along with the corresponding standard deviations. The novel repository used 1024 actors for data collection. We used NVIDIA V100 GPU for this experiment.

A notable observation from these experiments is the variation in success rates associated with different numbers of parallel actors. We hypothesize that this discrepancy arises due to the increased diversity

of data supplied to the replay buffer as the number of independent parallel environments increases, leading to more varied experiences for each policy update. We conducted similar experiments using our method with varying numbers of parallel environments to further investigate. The results are presented on the right side of Fig. 9. This observation, while interesting, is beyond the scope of the current work and is proposed as an area for further investigation.

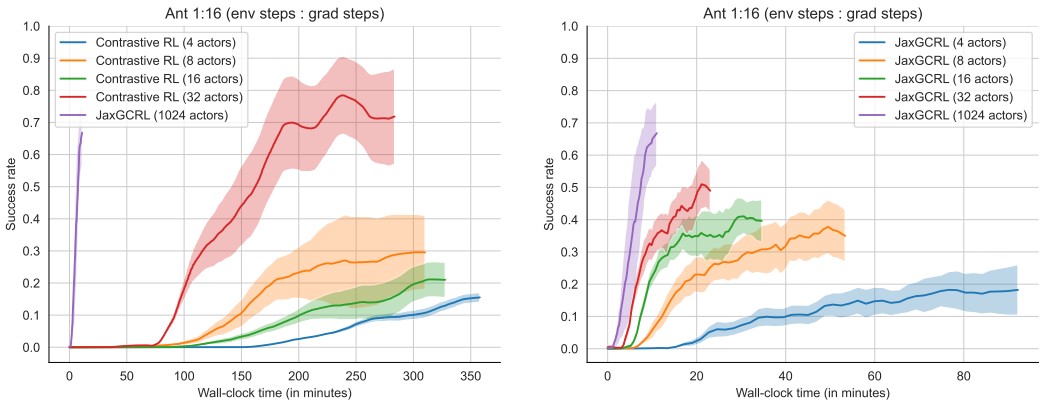

Figure 9: Speedup in ant environment for the 1:16 ratio of SGD steps : environment steps for different numbers of parallel actors.

## A.2 ENERGY FUNCTIONS AND CONTRASTIVE OBJECTIVES

The full list of evaluated energy functions:

$$f_{\phi,\psi,\cos}(s,a,g) = \frac{\langle \phi(s,a), \psi(g) \rangle}{\|\phi(s,a)\|_2 \|\psi(g)\|_2}, \tag{4}$$

$$f_{\phi,\psi,\text{dot}}(s,a,g) = \langle \phi(s,a), \psi(g) \rangle, \tag{5}$$

$$f_{\phi,\psi,L_1}(s,a,g) = -\|\phi(s,a) - \psi(g)\|_1, \tag{6}$$

$$f_{\phi,\psi,L_2}(s,a,g) = -\|\phi(s,a) - \psi(g)\|_2, \tag{7}$$

$$f_{\phi,\psi,L_2\ w\backslash o\ sqrt}(s,a,g) = -\|\phi(s,a) - \psi(g)\|_2^2. \tag{8}$$

The full list of tested contrastive objectives:

$$\mathcal{L}_{\text{InfoNCE-fwd}}(\mathcal{B}; \phi, \psi) = -\sum_{i=1}^{|\mathcal{B}|} \log\left(\frac{e^{f_{\phi,\psi}(s_i,a_i,g_i)}}{\sum_{j=1}^{K} e^{f_{\phi,\psi}(s_i,a_i,g_j)}}\right), \tag{9}$$

$$\mathcal{L}_{\text{InfoNCE-bwd}}(\mathcal{B}; \phi, \psi) = -\sum_{i=1}^{|\mathcal{B}|} \log\left(\frac{e^{f_{\phi,\psi}(s_i,a_i,g_i)}}{\sum_{j=1}^{K} e^{f_{\phi,\psi}(s_j,a_j,g_i)}}\right), \tag{10}$$

$$\mathcal{L}_{\text{InfoNCE-sym}}(\mathcal{B}; \phi, \psi) = \mathcal{L}_{\text{InfoNCE-fwd}}(\mathcal{B}; \phi, \psi) + \mathcal{L}_{\text{InfoNCE-bwd}}(\mathcal{B}; \phi, \psi), \tag{11}$$

$$\mathcal{L}_{\text{FlatNCE-fwd}}(\mathcal{B}; \phi, \psi) = -\sum_{i=1}^{|\mathcal{B}|} \log\left(\frac{\sum_{j=1}^{|\mathcal{B}|} e^{f_{\phi,\psi}(s_i,a_i,g_j)-f_{\phi,\psi}(s_i,a_i,g_i)}}{\text{detach}\left[\sum_{j=1}^{|\mathcal{B}|} e^{f_{\phi,\psi}(s_i,a_i,g_j)-f_{\phi,\psi}(s_i,a_i,g_i)}\right]}\right), \tag{12}$$

$$\mathcal{L}_{\text{FlatNCE-bwd}}(\mathcal{B}; \phi, \psi) = -\sum_{i=1}^{|\mathcal{B}|} \log\left(\frac{\sum_{j=1}^{|\mathcal{B}|} e^{f_{\phi,\psi}(s_j,a_j,g_i)-f_{\phi,\psi}(s_i,a_i,g_i)}}{\text{detach}\left[\sum_{j=1}^{|\mathcal{B}|} e^{f_{\phi,\psi}(s_j,a_j,g_i)-f_{\phi,\psi}(s_i,a_i,g_i)}\right]}\right), \tag{13}$$

$$\mathcal{L}_{\text{FB}}(\mathcal{B}; \phi, \psi) = -\sum_{i=1}^{|\mathcal{B}|} \left(e^{f_{\phi,\psi}(s_i,a_i,g_i)}\right) + \frac{1}{2(|\mathcal{B}|-1)} \sum_{j=1,j\neq i}^{K} \left(e^{f_{\phi,\psi}(s_i,a_i,g_j)}\right)^2, \tag{14}$$

$$\mathcal{L}_{\text{DPO}}(\mathcal{B}; \phi, \psi) = -\sum_{i=1}^{|\mathcal{B}|} \sum_{j=1}^{|\mathcal{B}|} \log \sigma\left[f_{\phi,\psi}(s_i,a_i,g_i) - f_{\phi,\psi}(s_i,a_i,g_j)\right], \tag{15}$$

$$\mathcal{L}_{\text{IPO}}(\mathcal{B}; \phi, \psi) = \sum_{i=1}^{|\mathcal{B}|} \sum_{j=1}^{|\mathcal{B}|} \left[\left(f_{\phi,\psi}(s_i,a_i,g_i) - f_{\phi,\psi}(s_i,a_i,g_j)\right) - 1\right]^2, \tag{16}$$

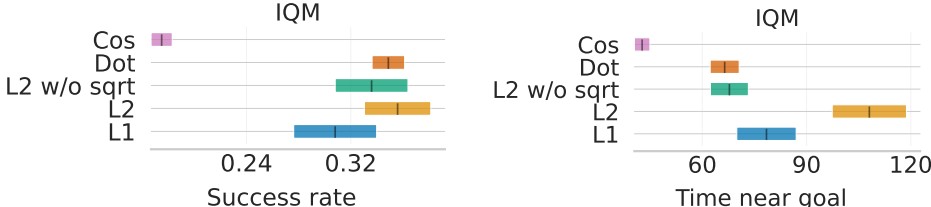

Figure 10: **Energy functions influence CRL performance metrics in multiple ways.** Success rate (left) and time near goal (right) results for 5 energy functions. IQMs indicate better performance of p-norms and dot product as energy functions over cosine similarity for CRL. Interestingly, L2 results in a much higher time near goal than other energy functions. Results averaged over five seeds and plotted with one standard error.

$$\mathcal{L}_{\text{SPPO}}(\mathcal{B}; \phi, \psi) = \sum_{i=1}^{|\mathcal{B}|} \sum_{j=1}^{|\mathcal{B}|} \left[ f_{\phi,\psi}(s_i, a_i, g_i) - 1 \right]^2 + \left[ f_{\phi,\psi}(s_i, a_i, g_j) + 1 \right]^2, \quad (17)$$

where we have highlighted the indices corresponding to positive and negative samples for clarity.

The last three of those losses, that is DPO, IPO, and SPPO were *inspired* by the structure of losses in the Preference Optimization domain. Unlike in the losses from InfoNCE family, here the samples are compared in pairs.

The DPO loss simply drives the difference between scores of positive and negative samples to be larger, and doesn't regularize those scores in any other way.

The IPO loss can be seen as a restriction of DPO, where the scores are regularized to always have a difference of one between a positive and negative sample pairs.

The SPPO loss restricts this even further, and regularizes the scores to be equal to one for positive samples and negative one for negative samples.

## A.3 ENERGY FUNCTIONS RESULTS

The performance of contrastive learning is sensitive to energy function choice (Sohn, 2016). This section aims to understand how different energy functions impact CRL performance. In particular, we evaluate five energy functions: L1, L2, L2 w/o sqrt, dot product and cosine. For every energy function, we use symmetric InfoNCE as a contrastive objective, with a 0.1 logsumexp penalty coefficient.

In Fig. 10, we report the performance of every energy function for four different ant environments and pusher. We find that p-norms and dot-product significantly outperform cosine similarity. Additionally, removing the root square from the L2 norm (L2 w/o sqrt) results in performance degradation, especially regarding time near goal. This modification makes the energy function no longer abide by the triangle inequality, which, as pointed out by (Myers et al., 2024b), is desirable for temporal contrastive features. Results per environment are reported in Fig. 11.

Fig. 11 shows the results of different energy functions per environment, success rate and time near goal. Clearly, no single energy function performs well in all the tested environments, as, for instance, L1 and L2, which perform well in Ant environments, work poorly in Pusher. In addition, we observed high variability in every configuration performance, as indicated by relatively wide standard errors.

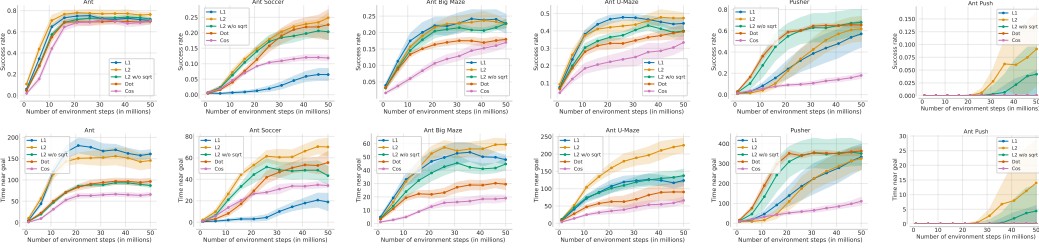

Figure 11: Success rate (top) and time near goal (bottom) results in different energy functions and environments. Best performing energy function varies across environments.

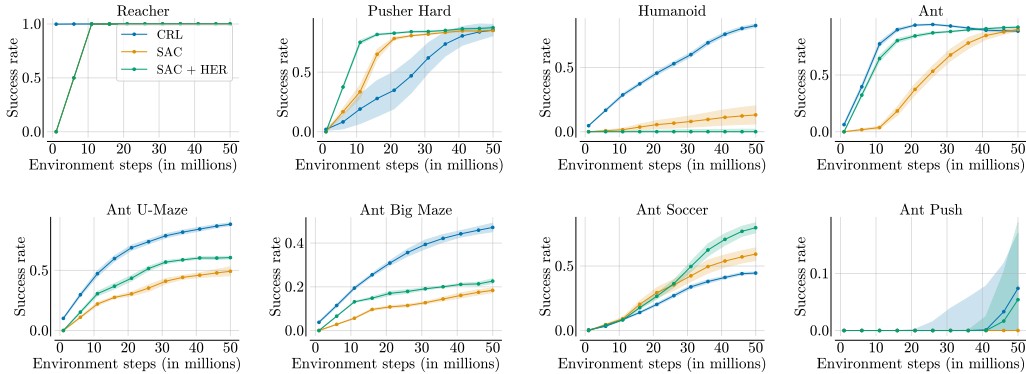

Figure 12: **Baseline results in JaxGCRL benchmark.** Success rates for each of our benchmark environments using bigger architecture. Reported as 10 seeds and standard error.

## A.4 Benchmark performance

In this section, report additional results for more advanced architectures on the benchmark environment. In particular, we report the results for architecture with 4 hidden layers of size 1024 and Layer Normalization in Fig. 12. Unsurprisingly, the performance is significantly better in most environments, particularly for Humanoid. This suggests that a larger architecture is needed to effectively manage the high-dimensional state and action spaces involved in this environment.

## A.5 Hindsight experience replay details

In HER, we relabel, on average, $50\%$ of goals with states achieved at the end of the rollout. The humanoid environment was trained on goals sampled from a distance in the range $[1.0, 5.0]$ meters and evaluated on goals sampled from a distance $5.0$ meters. All other environments were trained and evaluated on identical environments.

We observe poor performance for SAC+HER in the Pusher environment as a result of HER generating a "successful" experience, which is trivial. In particular, the goal in the pusher environment is the desired location of the puck, which is different from its initial position, so that agent should *push* the puck to that position. Relabeling the goal to the final location of the puck reached at the end of the episode often results in just changing the goal to the puck's initial position. This happens because, during the early stages of training, the random policy usually doesn't interact with the puck.

## A.6 Scaled-up CRL architecture in data-rich setting

In Figures 13 and 14, we report success rates and time near goal for scaled-up CRL agent with architecture consisting of 4 layers with 1024 neurons per layer and layer norm. We find that these architectures can increase the fraction of trials where the agent reaches the goal at least once, but they do not enable the agent to stabilise around the goal (e.g., on 7 tasks, the best agent spends less than $50\%$ of an episode at the goal).

When visualizing the rollouts, we observe that the Humanoid agent falls immediately after reaching the goal state, and the Ant Soccer agent struggles to recover when it pushes the ball too far away. The Humanoid merely "flings" itself toward the goal, while the optimal policy would involve running to the goal and remaining there. This inability to stabilize around the goal suggests that the agent is not effectively optimizing the actor's objective, pointing to a potential area for further research.

## B Technical details

`JaxGCRL` is a fast implementation of state-based self-supervised reinforcement learning algorithms and a new benchmark of GPU-accelerated environments. Our implementation leverages the power of GPU-accelerated simulators (BRAX and MuJoCo MJX) (Freeman et al., 2021; Todorov et al., 2012)

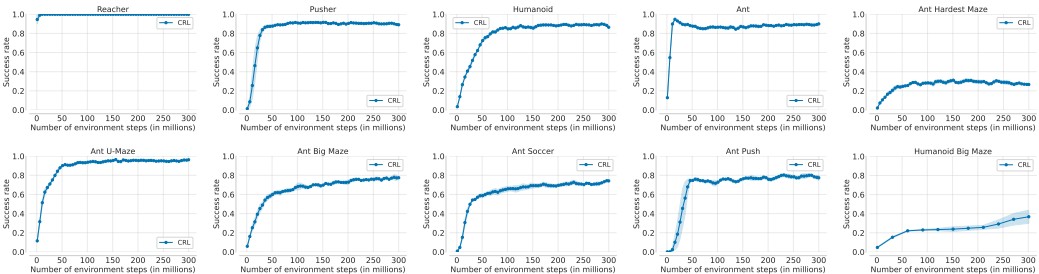

Figure 13: **CRL with big architecture success rates in data-rich setting.**

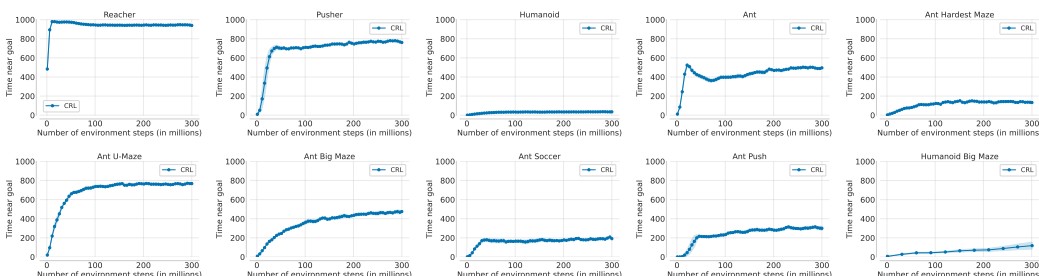

Figure 14: **CRL with big architecture time near goal in data-rich setting.**

to reduce the time required for data collection and training, allowing researchers to run extensive experiments in a fraction of the time previously needed. The bottleneck of former self-supervised RL implementations was twofold. Firstly, data collection was executed on many CPU threads, as a single thread was often used for a single actor. This reduced the number of possible parallel workers, as only high-compute servers could run hundreds of parallel actors. Secondly, the necessity of data migration between CPU (data collection) and GPU (training) posed additional overhead. These two problems are mitigated by fully JIT-compiled algorithm implementation and execution of all environments and replay buffer operations directly on the GPU.

Notably, our implementation uses only one CPU thread and has low RAM usage as all the operations, including those on the replay buffer, are computed on GPU. It's important to note that the BRAX physics simulator is not exactly the same as the original MuJoCo simulator, so the performance numbers reported here are slightly different from those in prior work. All methods and baselines we report are run on the same BRAX simulator.

### B.1 ENVIRONMENT DETAILS

In each of the environments, there are a number of parameters that can change the learning process. A non-exhaustive list of such details for each environment is presented below.

### B.2 BENCHMARK DETAILS

Our experiments use `JaxGCRL` suite of simulated environments described in Section 4.2. We evaluate algorithms in an online setting, with a UTD ratio 1:16 for CRL, TD3, TD3+HER, SAC, SAC+HER, and $1:5$ for PPO. We use a batch size of $256$ and a discount factor of $0.99$ for all methods except PPO, for which we use a discount factor of $0.97$. For every environment, we sample evaluation goals from the same distribution as training ones and use a replay buffer of size 10M for CRL, TD3, TD3+HER, SAC, and SAC+HER. We use $1024$ parallel environments for all methods except for PPO, where we use $4096$ parallel environments to collect data. All experiments are conducted for $50$ million environment steps.

---

[1]in 3 dimensions

Table 1: Environments details

| Environment | Goal distance | Termination | Brax pipeline | Goal sampling |
|---|---|---|---|---|
| Reacher | 0.05 | No | Spring | Disc with radius sampled from $[0.0, 0.2]$ |
| Half-Cheetah | 0.5 | No | MJX | Fixed goal location |
| Pusher Easy | 0.1 | No | Generalized | x coordinate sampled from $[-0.55, -0.25]$, y coordinate sampled from $[-0.2, 0.2]$ |
| Pusher Hard | 0.1 | No | Generalized | x coordinate sampled from $[-0.65, 0.35]$, y coordinate sampled from $[-0.55, 0.45]$ |
| Humanoid | $0.5$[1] | Yes | Spring | Disc with radius sampled from $[1.0, 5.0]$ |
| Ant | 0.5 | Yes | Spring | Circle with radius 10.0 |
| Ant Maze | 0.5 | Yes | Spring | Maze specific |
| Ant Soccer | 0.5 | Yes | Spring | Circle with radius 5.0 |
| Ant Push | 0.5 | Yes | MJX | Two possible locations with uniform noise added |

Table 2: Hyperparameters

| Hyperparameter | Value |
|---|---|
| num_timesteps | 50,000,000 |
| max_replay_size | 10,000 |
| min_replay_size | 1,000 |
| episode_length | 1,000 |
| discounting | 0.99 |
| num_envs | 1024 (512 for humanoid) |
| batch_size | 256 |
| multiplier_num_sgd_steps | 1 |
| action_repeat | 1 |
| unroll_length | 62 |
| policy_lr | 6e-4 |
| critic_lr | 3e-4 |
| contrastive_loss_function | symmetric_infonce |
| energy_function | L2 |
| logsumexp_penalty | 0.1 |
| hidden layers (for both encoders and actor) | [256,256] |
| representation dimension | 64 |

## B.3 BENCHMARK PARAMETERS

The parameters used for benchmarking experiments can be found in Table 2.
`min_replay_size` is a parameter that controls how many transitions **per environment** should be gathered to prefill the replay buffer.
`max_replay_size` is a parameter that controls how many transitions are maximally stored in replay buffer **per environment**.

## C RANDOM GOALS

Our loss in Eq. (3) differs from the original CRL algorithm (Eysenbach et al., 2022) by sampling goals from the same trajectories as states during policy extraction, rather than random goals from the replay buffer. Mathematically, we can generalize Eq. (3) to account for either of these strategies

by adding a hyperparameter $\alpha$ to the loss controlling the degree of random goal sampling during training.

$$\max_{\theta} \quad (1 - \alpha) \cdot \mathbb{E}_{p(s,a)p(g|s,a)\pi_{\theta}(a'|s,g)} \left[ f_{\phi,\psi}(s, a', g) \right]$$

$$+ \alpha \cdot \mathbb{E}_{p(s,a)p(g)\pi_{\theta}(a'|s,g)} \left[ f_{\phi,\psi}(s, a', g) \right]$$

The hyperparameter $\alpha$ controls the rate of counterfactual goal learning, where the policy is updated based on the critic's evaluation of goals that did not actually occur in the trajectory. We find that taking, $\alpha = 0$ (i.e., no random goal sampling) leads to better performance, and suggest using the policy loss in Eq. (3) for training contrastive RL methods.

# D PROOFS

## D.1 Q-FUNCTION IS PROBABILITY

This proof follows closely the one presented in Eysenbach et al. (2022).

We want to relate the Q-function to discounted state visitation distribution: $Q_g^{\pi}(s, a) = p_{\gamma}^{\pi}(g \mid s, a)$. The Q-function is usually defined in terms of rewards:

$$Q_g^{\pi}(s, a) \triangleq \mathbb{E}_{\pi(\cdot|g)} \left[ \sum_{t=0}^{\infty} \gamma^t r_g(s_t, a_t) \mid {}^{s_0=s,}_{a_0=a} \right]. \tag{18}$$

We will define rewards conditioned with goal $g$ as:

$$r_g(s, a) \triangleq \begin{cases} (1 - \gamma)\big(p(s_0 = g) + \gamma p(s_1 = g \mid s_0, a_0)\big), & t = 0 \\ (1 - \gamma)\gamma p(s_{t+1} = g \mid s_t, a_t), & t > 0. \end{cases} \tag{19}$$

Lastly, we define discounted state visitation distribution:

$$p_{\gamma}^{\pi}(g) \triangleq (1 - \gamma) \sum_{t=0}^{\infty} \gamma^t p_t^{\pi}(g). \tag{20}$$

For $t > 0$, the term $p_t^{\pi}(g)$ is a probability of reaching the goal $g$ at timestep $t$ with policy conditioned on $g$, and thus:

$$p_t^{\pi}(g) = \mathbb{E}_{\pi(\cdot|g)} \big[ p_t(g \mid s_{t-1}, a_{t-1}) \big]$$
$$= \mathbb{E}_{\pi(\cdot|g)} \big[ p(s_t = g \mid s_{t-1}, a_{t-1}) \big].$$

On the second line, we have used the Markov property. We can now substitute this into Eq. (20):

$$p_{\gamma}^{\pi}(g) = (1 - \gamma) \sum_{t=0}^{\infty} \gamma^t p_t^{\pi}(g)$$

$$= (1 - \gamma) p_0^{\pi}(g) + (1 - \gamma) \sum_{t=1}^{\infty} \gamma^t \mathbb{E}_{\pi(\cdot|g)} \big[ p(s_t = g \mid s_{t-1}, a_{t-1}) \big]$$

$$= (1 - \gamma) p_0^{\pi}(g) + (1 - \gamma) \sum_{t=0}^{\infty} \gamma^{t+1} \mathbb{E}_{\pi(\cdot|g)} \big[ p(s_{t+1} = g \mid s_t, a_t) \big]$$

$$= \mathbb{E}_{\pi(\cdot|g)} \left[ (1 - \gamma) p(s_0 = g) + (1 - \gamma) \sum_{t=0}^{\infty} \gamma^{t+1} p(s_{t+1} = g \mid s_t, a_t) \right]$$

$$= \mathbb{E}_{\pi(\cdot|g)} \left[ \underbrace{(1 - \gamma) \left( p(s_0 = g) + \gamma p(s_1 = g \mid s_0, a_0) \right)}_{r_g(s_0, a_0)} + \sum_{t=1}^{\infty} \gamma^t \underbrace{(1 - \gamma)\gamma p(s_{t+1} = g \mid s_t, a_t)}_{r_g(s_t, a_t)} \right]$$

$$= \mathbb{E}_{\pi(\cdot|g)} \left[ \sum_{t=0}^{\infty} \gamma^t r_g(s_t, a_t) \right].$$

Thus for a set state-action pair $(s, a)$, we have:

$$p_\gamma^\pi(g \mid s, a) = \mathbb{E}_{\pi(\cdot|g)} \left[ \sum_{t=0}^{\infty} \gamma^t r_g(s_t, a_t) \mid {}^{s_0=s,}_{a_0=a} \right] = Q_g^\pi(s, a),$$

which relates the Q-function to discounted state visitation distribution.

## E   FAILED EXPERIMENTS

1. *Weight Decay:* Prior work (Nauman et al., 2024b) indicated that regularizing critic weights might improve learning stability. We did not observe significant upgrades in the CRL setup, perhaps due to a much lower ratio of updates per environment step. We tested this only for small architectures.
2. *Random Goals:* In prior implementation (Eysenbach et al., 2022) using random goals in actor loss resulted in higher performance. We did not observe that in our online setting.

## F   PSEUDOCODE

Pseudocode for the contrastive learning algorithms studied is presented in Algorithm 1.

---

**Algorithm 1** Contrastive Reinforcement Learning
---

1: **Input:** Contrastive loss $\mathcal{L}_{\text{Critic}}$, energy function $f$
2: Initialize $\phi, \psi, \pi$ and an empty replay buffer $\mathcal{D}$
3: **repeat**
4:    **in parallel over environments**
5:       Observe state $s$ and sample an action $a \sim \pi(s, g)$
6:       Execute $a$ in the environment
7:       Observe next state $s'$ and done signal $d$ to indicate whether $s'$ is terminal
8:       Append $(s, a, s')$ to current trajectory for this environment
9:       **if** $s'$ is terminal **then**
10:          Reset environment state and sample new goal
11:          Store current trajectory for this environment in $\mathcal{D}$
12:          Start new trajectory for this environment
13:    **for** $j = 1, \dots, \texttt{num\_updates}$ **do**
14:       Randomly sample (with discount) a batch $\mathcal{B}$ from $\mathcal{D}$ of state-action pairs and goals from their future
15:       Update critic:
         $(\phi, \psi) \leftarrow (\phi, \psi) - \alpha \nabla_{\phi,\psi} \left[ \mathcal{L}_{\text{Critic}}(\mathcal{B}; \phi, \psi) + \beta \mathcal{L}_{\text{logsumexp}}(\mathcal{B}, \phi, \psi) \right]$
16:       Update policy:
         $\pi \leftarrow \pi - \alpha \nabla_\pi \left[ \mathcal{L}_{\text{Actor}}(\mathcal{B}; \phi, \psi, \pi) \right]$
17: **until** convergence

---

