# OpenReview forum: "Accelerating Goal-Conditioned Reinforcement Learning Algorithms and Research"
_ICLR.cc/2025/Conference — ICLR 2025 Spotlight_

### Official Review · Reviewer_3DLK · 2024-10-31

**Soundness:** 3
**Presentation:** 3
**Contribution:** 2
**Rating:** 6
**Confidence:** 3

**Summary:**

The authors introduce JaxGCRL, a benchmark and framework for evaluating goal-conditioned RL algorithms based on contrastive learning.
They re-implement a number of goal-conditioned tasks from prior literature and evaluate their implementation on it.

They then evaluate the effect of different losses, more samples, and larger networks on their implementation. They demonstrate that their Jax-based implementation is significantly faster than previous libraries, accelerating future research.

**Strengths:**

The paper has several significant strengths.

- Although not novel, JAX implementations are to be commended. They improve research iteration speed significantly.
- The paper is very well written. The authors communicate their results clearly and unambiguously.
- The authors evaluate using the inter-quartile mean and bootstrapped confidence intervals. This is more sound than using learning curves etc.
- The authors provide a number of ablations and experiments that explain the performance of their implementation.

**Weaknesses:**

However, I have a number of issues with this paper, which is why I recommend rejection.

- The authors claim that their setting is challenging, but do not effectively demonstrate that this is the case. The authors demonstrate that by using a bigger network (1024 layer width and depth of 4) and layer norm, the performance significantly improves. They also run experiments where they train for significantly more interactions. However, as best I can tell (it is not always clear which network is used in which experiment), the authors never run their biggest, highest performing network for 300M steps on all the tasks. The authors do not pitch their work as focussing on sample efficiency, and therefore I am not sure why their evaluation framework should be compelling if the tasks can be solved by scaling up networks and using more samples. If the authors can provide a demonstration that this does not satisfactorily solve their benchmark, **I will raise my score**. However, without this demonstration, I do not believe that the experimental insights and JAX implementation are enough to warrant acceptance.
- I am confused about the experiments concerning the update-to-date ratio (UTD). Given a fixed step budget, doing fewer or more updates is a pure trade-off. You can do fewer, less noisy updates, or do more, noisier updates. This occurs all over RL, for example when choosing the number of parallel environments to use in PPO. I am not sure why a high or low number of updates would be beneficial, or this quantity would be interesting to examine.

I also have a number of more minor points:
- The authors claim that they cannot directly compare brax and mujoco because brax uses a different physics engine, but the MuJoCo physics engine has been available in brax for a while now [1] -- what exactly is the issue here?
- The discussion of related work on jax-based environments is missing some work. Gymnax [2] and PureJaxRL [3], both were important landmarks in the use of and benefits of JAX in RL and warrant inclusion.
- The authors should probably rephrase line 117, which begins with "In addition to reducing CPU-GPU data-transfer overhead...". While implementing an environment in JAX *does* do this, there are also significant other factors such as JIT compilation and the resulting operator fusion and the ability to use more vectorised environments than a typical CPU + pytorch approach that lead to the significant speedups.
- A number of the papers listed in the appendix have incorrect citations or are missing authors.
- Line 1032 in the appendix contains a typo (lenght -> length)

**Questions:**

See weaknesses.

[1] Brax documentation. https://github.com/google/brax?tab=readme-ov-file#one-api-four-pipelines

[2] Gymnax: A JAX-based Reinforcement Learning Library. Robert Lange. https://github.com/RobertTLange/gymnax

[3] Lu, Chris, et al. "Discovered policy optimisation." Advances in Neural Information Processing Systems 35 (2022): 16455-16468. https://github.com/luchris429/purejaxrl

---

> ### Author Response · Authors · 2024-11-17
>
> We thank the reviewer for their time and feedback on the work. It seems like the reviewer's main concern is that the tasks in the benchmark may not be difficult enough. To test this hypothesis, we trained our biggest network (1024 layer width and depth of 4 and layer norm) for 300M steps on all the tasks. This scaled-up model achieves over a 50% [success rate](https://anonymous.4open.science/r/paperJaxGCRL-EFEB/success_rate.png) on the proposed tasks. However, it [struggles to stabilise at the goal](https://anonymous.4open.science/r/paperJaxGCRL-EFEB/time_near_goal.png); for instance, on five tasks, the best-performing agent spends less than 50% of an episode at the goal. *Does this address the reviewer's concern about the tasks not being difficult enough?* If not, we are happy to run additional experiments or add additional environments. One example of such an environment is the Ant Hard Maze, where the scaled-up CRL model fails to achieve a success rate of 50%.
>
> > I am confused about the experiments concerning the update-to-date ratio (UTD). Given a fixed step budget, doing fewer or more updates is a pure trade-off. You can do fewer, less noisy updates, or do more, noisier updates. This occurs all over RL, for example when choosing the number of parallel environments to use in PPO. I am not sure why a high or low number of updates would be beneficial, or this quantity would be interesting to examine.
>
> We include this ablation experiment because prior work [1-3] has found that UTD can be important for certain RL algorithms. Unlike prior work, we found that CRL with a low UTD works **better** on several tasks.
>
> > The authors claim that they cannot directly compare brax and mujoco because brax uses a different physics engine, but the MuJoCo physics engine has been available in brax for a while now [1] -- what exactly is the issue here?
>
> There are some unfortunate naming conventions, where "Brax" refers to a physics _library_ that now includes several different physics _engines_: Positional, Spring, Generalised, and MJX for a while. The "original" Mujoco ant environments used the MuJoCo physics engine. In contrast, the Brax `Ant` environments use the Spring physics engine by default, so performance on one benchmark isn't the same as performance on the other.
> We would also like to note that not all Brax environments currently support MJX [5]. We encountered issues running Ant or Pusher with the MJX backend out-of-the-box. Additional tuning of the MJX physics engine may be required in these environments.
>
> > - The discussion of related work on jax-based environments is missing some work. Gymnax [2] and PureJaxRL [3], both were important landmarks in the use of and benefits of JAX in RL and warrant inclusion.
> > - The authors should probably rephrase line 117, which begins with "In addition to reducing CPU-GPU data-transfer overhead...". > - > - While implementing an environment in JAX does do this, there are also significant other factors such as JIT compilation and the resulting operator fusion and the ability to use more vectorised environments than a typical CPU + pytorch approach that lead to the significant speedups.
> > - A number of the papers listed in the appendix have incorrect citations or are missing authors.
> > - Line 1032 in the appendix contains a typo (lenght -> length)
>
>
> We appreciate the reviewer for pointing out these issues. We have fixed all of them in the [new manuscript version](https://anonymous.4open.science/r/paperJaxGCRL-EFEB/Accelerating_Goal_Conditioned_RL_Algorithms_and_Research_rebuttal.pdf).
>
>
> [1] D’Oro, P., Schwarzer, M., Nikishin, E., Bacon, P.-L., Bellemare, M. G., & Courville, A. (2022, September 29). Sample-Efficient Reinforcement Learning by Breaking the Replay Ratio Barrier. The Eleventh International Conference on Learning Representations. https://openreview.net/forum?id=OpC-9aBBVJe
>
> [2] Schwarzer, M., Obando-Ceron, J., Courville, A., Bellemare, M., Agarwal, R., & Castro, P. S. (2023, June 9). Bigger, Better, Faster: Human-level Atari with human-level efficiency. http://arxiv.org/abs/2305.19452
>
> [3] Nauman, M., Ostaszewski, M., Jankowski, K., Miłoś, P., & Cygan, M. (2024, May 25). Bigger, Regularized, Optimistic: Scaling for compute and sample-efficient continuous control. http://arxiv.org/abs/2405.16158
>
> [4] Spring Backend https://github.com/google/brax?tab=readme-ov-file#one-api-four-pipelines:~:text=and%20collision%20constraints.-,Spring,-provides%20fast%20and
>
> [5] MJX support https://github.com/google/brax/discussions/409#:~:text=We%20will%20work%20to%20port%20MJX%20into%20Brax%20as%20another%20physics%20pipeline

---

> ### Comment · Reviewer_3DLK · 2024-11-20
> **Thank you for your response**
>
> I'd like to thank the authors for their additional experiments and response to my questions. I appreciate the running of the experiments with the larger model and thank the authors for their efforts.
>
> I'm not sure that the current experiments do address my concern -- if this is to be a meaningful benchmark, the there should be significant headroom on some of the tasks, so that progress can be measured without resorting to the ceiling effect. As I see it the success rates are all quite high still. I understand that the agents fail to stabilise near the goal, but I am unsure why this is a significant problem? Does this indicate that the policies are very far from optimal? Is it possible to be at the goal for a large proportion of the time? I could see that when, for example, solving a maze, it would take a long time to find the goal, and therefore time near goal is likely to be quite small. To my knowledge all that is 'rewarded' here is achieving the goal right? If you were optimising for time near the goal it would be a valid concern to complain about insufficient time near the goal, but you are not as far as I understand it -- just reaching it in the first place.
>
> Adding a couple of harder environments with worse success rates would alleviate my concerns -- I would then raise my score, but if the current policies can also be shown to be clearly far from optimal, that would also provide sufficient evidence
>
> On the point of the update-to-data ratio, I do not agree with the authors' understanding of the prior work. The argument in those papers (to my knowledge) is not that a high update-to-data ratio is *always good*, but that prior work had been *forced* to use a low update to data ratio (because otherwise performance collapses or isn't as good etc.), thereby harming their achievable *sample efficiency*. Therefore by coming up with a method (presented in those papers) that *can achieve* high update-to-data ratio, they should expect gains in sample efficiency. Those papers are also specific to off-policy Q-learning algorithms to my knowledge. Off-policy methods should be able to achieve a higher update-to-data ratio. An on-policy algorithm (such as PPO) is clearly going to collapse with a high update-to-data ratio. Given that your method seems (at least to me) to be on-policy, it is not clear to me why it would be expected that such an update-to-data result would hold, and therefore why they are in the paper.
>
> I'd also like to note my disagreement with the documentation requirements laid out by reviewer MvLy. While I think that documentation is important, and all the steps suggested by the reviewer would be helpful, I think that the suggestions are clearly above the required level of documentation for previous JAX frameworks accepted at top ML conferences [1, 2]
>
> [1] Matthews et al. Craftax: A Lightning-Fast Benchmark for Open-Ended Reinforcement Learning
> [2] Rutherford et al. JaxMARL: Multi-Agent RL Environments in JAX

---

> > ### Author Response · Authors · 2024-11-21
> >
> > Dear Reviewer,
> >
> > Thanks for continuing the discussion. To address your concern regarding the benchmark, we are introducing two challenging environments: Ant Hardest Maze and Humanoid Big Maze, where the success rates are ~30% and ~40%, respectively (see Figures 13 and 14).
> >
> > We also want to clarify that the agent is optimized for the time near the goal (see [`actor_loss` function](https://anonymous.4open.science/r/JaxGCRL-2316/src/losses.py)). According to the original [contrastive RL paper](https://arxiv.org/pdf/2206.07568):
> >
> > > "Intuitively, this objective corresponds to sampling a goal s_g and then optimizing the policy to go to that goal and stay there." (bottom of page 3)
> >
> > Thus, the agent's failure to stabilise around the goal does mean that the agent is not doing a great job optimizing the objective. For example, we can look at a video of the [humanoid task](https://anonymous.4open.science/r/paperJaxGCRL-EFEB/README.md): the humanoid simply "flings" itself at the goal, whereas the optimal policy would be to run towards the goal and stand there. We know that methods with dense rewards can learn this behavior, but are unaware of goal-conditioned methods (i.e., given just a goal and no dense rewards) that can learn this behavior. We have revised Sec A.6 to clarify this.
> >
> > > On the point of the update-to-data ratio, I do not agree with the authors' understanding of the prior work. The argument in those papers (to my knowledge) is not that a high update-to-data ratio is always good, but that prior work had been forced to use a low update to data ratio (because otherwise performance collapses or isn't as good etc.), thereby harming their achievable sample efficiency. Therefore by coming up with a method (presented in those papers) that can achieve high update-to-data ratio, they should expect gains in sample efficiency. Those papers are also specific to off-policy Q-learning algorithms to my knowledge. Off-policy methods should be able to achieve a higher update-to-data ratio. An on-policy algorithm (such as PPO) is clearly going to collapse with a high update-to-data ratio. Given that your method seems (at least to me) to be on-policy, it is not clear to me why it would be expected that such an update-to-data result would hold, and therefore why they are in the paper.
> >
> > Thank you for clarifying these points. Based on your feedback, we have revised Section 5.6. Additionally, we have updated the description of our UTD experiment to highlight that it serves as an example of a typically computationally expensive experiment that runs significantly faster using our code. The aim of these experiments is not to make a claim about whether the phenomenon we observe is the same/different from those observed in off-policy methods.
> >
> > Do these further revisions fully address the reviewer's concerns with the paper? If not, we would be happy to run additional experiments and revise the paper further.
> >
> > Kind regards,
> >
> > Authors

---

> > > ### Comment · Reviewer_3DLK · 2024-11-21
> > > **Thank you for the response**
> > >
> > > I'd like to thank the authors for engaging with my concerns. The added environments are clearly difficult even for the largest network presented and I will therefore raise my score as promised.
> > >
> > > Thank you for clarifying the point around stabilisation around the goal and answering the rest of my questions.

---

### Official Review · Reviewer_fgPt · 2024-11-01

**Soundness:** 4
**Presentation:** 4
**Contribution:** 4
**Rating:** 8
**Confidence:** 4

**Summary:**

The paper provides a JIT-complied codebase with vectorized environments that can speed up the training and iterating new ideas on goal-conditioned reinforcement learning problems.
In additional, it provides a stable baseline algorithm for the goal-conditioned reinforcement learning problems that's benchmarked in the 8 diverse continuous environments.

**Strengths:**

* The JaxGCRL codebase is significantly faster than the original codebase.
* The proposed baseline consistently outperform the counterpart in all 8 environments, demonstrating the stableness from simple to complex environments.
* The performance of different design choice is extensively tested and the result metric is easy to interpret.

**Weaknesses:**

The paper mentions it leverages the power of GPU-accelerated simulators, but by comparing against the brax training code under https://github.com/google/brax/tree/main/brax/training, there are some similarities for the training code as well, and it's not mentioned in the paper.

**Questions:**

* In Sec 5.3, why is the contrastive objective only evaluated on part of the 8 environments? Similar question in sec 5.6 for examining different UTD ratios.
* In Fig 1. Are the num_actors same for the JaxGCRL and CRL?
* How do you define if the agent is within goal's proximity?

---

> ### Author Response · Authors · 2024-11-17
>
> We thank the reviewer for their time and efforts in working on the review and for their kind words on our work.
>
> >The paper mentions it leverages the power of GPU-accelerated simulators, but by comparing against the brax training code under https://github.com/google/brax/tree/main/brax/training, there are some similarities for the training code as well, and it's not mentioned in the paper.
>
> We thank the reviewer for this comment. Our work does build upon Brax, extending the prior work to develop a new benchmark for goal-conditioned RL tasks. In contrast, Brax focuses on single-reward tasks. We accordingly modify Section 5.1 Experimental Setup to indicate that JaxGCRL implementation is based on SAC implementation from Brax.
>
> > In Sec 5.3, why is the contrastive objective only evaluated on part of the 8 environments? Similar question in sec 5.6 for examining different UTD ratios.
>
> Finite compute resources: Figures in these sections are already the result of 800+ training runs. In the updated [manuscript](https://anonymous.4open.science/r/paperJaxGCRL-EFEB/Accelerating_Goal_Conditioned_RL_Algorithms_and_Research_rebuttal.pdf), we have included Ant Push in the energy function experiments. In the coming days, we will also add this environment to the contrastive objective and UTD ratio sections.
>
> > In Fig 1. Are the num_actors same for the JaxGCRL and CRL?
>
> No. For a fair comparison, we tuned the `num_actors` for both JaxGCRL and CRL and reported the best results for both.
>
> > How do you define if the agent is within goal's proximity?
>
> We have revised Section 5.1 (Experimental Setup) to direct readers to Table 1, which provides detailed definitions of proximity for each task.

---

> > ### Comment · Reviewer_fgPt · 2024-11-23
> >
> > Thank author for addressing my comments and adding more experiments.
> >
> > Overall the paper is in good shape, I would recommend acceptance.
> >
> > Nit: ill-formatted citation at line 33

---

### Official Review · Reviewer_j3bj · 2024-11-02

**Soundness:** 3
**Presentation:** 3
**Contribution:** 3
**Rating:** 8
**Confidence:** 4

**Summary:**

This paper introduces JaxGCRL, a codebase that contains environments and a scalable goal-conditioned RL algorithm, all implemented in JAX. This allows researchers to train GC agents much faster than before, making these experiments more accessible. This work also analyses several design decisions of contrastive RL algorithms, enabled by a fast simulator & algorithm implementation.

**Strengths:**

- Adding even more environments and algorithms to the JAX ecosystem is great, especially for goal-conditioned RL which is lacking in this space.
- The proof-of-concept experiments demonstrate what this library can allow, namely, more thorough investigation of design decisions in goal-conditioned RL
- The writing and motivation is quite clear.

**Weaknesses:**

- I can't see any major weaknesses, apart from the limited number of environments, although 8 is pretty respectable.

**Questions:**

- What is your support plan going forward with JaxGCRL, are you planning on adding new environments or algorithms?
- It seems like JaxGCRL is very much focused on brax-type environments, is there a part of goal conditioned RL research that potentially focuses rather on discrete action environments that you are leaving out?
- What about other non-contrastive GCRL algorithms? Are you planning on adding support for those?
	- Relatedly, how easy would it be for someone else to implement a new GCRL algorithm to fit within your framework?
	- And how easy is it to add another goal conditioned environment, based on an existing JAX environment? For instance, minigrid or craftax or xland minigrid, etc?
- In the maze, for instance, can you dynamically, within the JIT, change the maze layout, or does it have to be statically known at compile time?
- Is there an easy way to transfer a JaxGCRL agent to existing environments?

---

> ### Author Response · Authors · 2024-11-17
>
> We thank the reviewer for their time and efforts in working on the review and for their kind words on our work.
>
> > I can't see any major weaknesses, apart from the limited number of environments, although 8 is pretty respectable.
>
> Since the initial submission, we have added 5 new tasks that involve robotic manipulation and one harder ant maze environment.
>
> > What is your support plan going forward with JaxGCRL, are you planning on adding new environments or algorithms?
>
> Our support plan assumes active maintenance of JaxGCRL in the following months, focusing on gaining the necessary visibility and contributors. We realise that JaxGCRL will only provide value to the community if it becomes the out-of-the-shelf solution for custom GCRL experiments. To improve usability, we enhanced the repository by adding detailed documentation, multiple examples, and an updated `README.md` file.
>
> > It seems like JaxGCRL is very much focused on brax-type environments, is there a part of goal conditioned RL research that potentially focuses rather on discrete action environments that you are leaving out?
>
> Yes, there's a good bit of prior work on GCRL in settings with discrete actions [1-4]. However, our focus is on the likewise well-studied problem of GCRL in settings with continuous actions [5,6].
>
>
> > What about other non-contrastive GCRL algorithms? Are you planning on adding support for those?
> Relatedly, how easy would it be for someone else to implement a new GCRL algorithm to fit within your framework?
> And how easy is it to add another goal conditioned environment, based on an existing JAX environment? For instance, minigrid or craftax or xland minigrid, etc?
>
> We have added additional non-contrastive GCRL algorithms, including PPO and TD3 ([Figure 3](https://anonymous.4open.science/r/paperJaxGCRL-EFEB/Accelerating_Goal_Conditioned_RL_Algorithms_and_Research_rebuttal.pdf)). Adding a new algorithm is relatively easy because changes should concern mostly `networks.py` and `losses.py` files. Adding a novel, already working, MJX-based environment to JaxGCRL is also straightforward. We modified `README.md` to describe this process in greater detail.
>
> > In the maze, for instance, can you dynamically, within the JIT, change the maze layout, or does it have to be statically known at compile time?
>
> Currently, dynamically changing the maze layout is not supported, and the layout must be provided at the beginning of the experiment.
>
> > Is there an easy way to transfer a JaxGCRL agent to existing environments?
>
> We would like to further inquire about the specific environments the reviewer refers to. It is easy to transfer the JaxGCRL agent to other MJX-based environments. Environments created for MuJoCo can be, in most cases, adapted for MJX (as explained in the Feature Parity document [7]). In some situations, migration can be challenging since it requires rewriting the logic of the environment in JAX and thus requires some technical knowledge.
>
> [1] Chevalier-Boisvert, M., Dai, B., Towers, M., Lazcano, R. de, Willems, L., Lahlou, S., Pal, S., Castro, P. S., & Terry, J. (2023, June 24). Minigrid & Miniworld: Modular & Customizable Reinforcement Learning Environments for Goal-Oriented Tasks. http://arxiv.org/abs/2306.13831
>
> [2] Hoang, C., Sohn, S., Choi, J., Carvalho, W., & Lee, H. (2021, November 18). Successor Feature Landmarks for Long-Horizon Goal-Conditioned Reinforcement Learning. http://arxiv.org/abs/2111.09858
>
> [3] Nikulin, A., Kurenkov, V., Zisman, I., Agarkov, A., Sinii, V., & Kolesnikov, S. (2024, February 6). XLand-MiniGrid: Scalable Meta-Reinforcement Learning Environments in JAX. http://arxiv.org/abs/2312.12044
>
> [4] Liu, M., Zhu, M., & Zhang, W. (2022, September 2). Goal-Conditioned Reinforcement Learning: Problems and Solutions. http://arxiv.org/abs/2201.08299
>
> [5]  Chane-Sane, E., Schmid, C., & Laptev, I. (2021, July 1). Goal-Conditioned Reinforcement Learning with Imagined Subgoals. http://arxiv.org/abs/2107.00541
>
> [6] Nasiriany, S., Pong, V. H., Lin, S., & Levine, S. (2019, November 19). Planning with Goal-Conditioned Policies. http://arxiv.org/abs/1911.08453
>
> [7] https://mujoco.readthedocs.io/en/3.0.1/mjx.html#feature-parity

---

> > ### Comment · Reviewer_j3bj · 2024-11-18
> >
> > Thank you for your response! I do appreciate the addition of the new environments, new algorithms, and the new docs.
> >
> >
> > > We would like to further inquire about the specific environments the reviewer refers to.
> >
> >
> > What I had in mind here was, for instance, doing fast training using JaxGCRL, and then using that trained agent in current, non-jaxified environments (e.g. what researchers were using before). I guess the response about MJX vs Mujoco does answer this, in that transferring agents directly may not result in amazing performance due to the slight dynamics differences between the two physics engines. Disregarding that, however, how involved would the coding/porting need to be if I wanted to export a JaxGCRL agent (say for MJX ant maze) and deploy it on a non-jax, gymnasium-based ant-maze environment?

---

> > > ### Author Response · Authors · 2024-11-19
> > >
> > > Taking a trained agent in JaxGCRL agent (say for MJX ant maze) and deploying it in a non-jax, gymnasium-based ant-maze environment is straightforward to implement. Running the agent in the Gymnasium-based environment would only require loading the models trained with JaxGCRL. In fact, in the context of single-task RL, prior work has already done effectively the same thing; Humanoid Bench [1] first trains a one-hand reaching policy using massively parallelized PPO with MuJoCo MJX, and later adapts that policy to more challenging tasks (simulated in classical MuJoCo) with a higher number of potential collisions.
> > >
> > > [1] [https://humanoid-bench.github.io/](https://humanoid-bench.github.io/)

---

> > > > ### Comment · Reviewer_j3bj · 2024-11-23
> > > >
> > > > Great, thank you for elaborating. I will keep my score of 8, as I believe this paper should be accepted.

---

### Official Review · Reviewer_MvLy · 2024-11-03

**Soundness:** 3
**Presentation:** 3
**Contribution:** 1
**Rating:** 8
**Confidence:** 4

**Summary:**

The authors propose a new library for goal conditioned reinforcement learning (GCRL). Unlike prior work, their method runs end to end on the GPU, making training faster. They implement 8 environments in JAX, as well a few algorithms and various objectives. Then, they evaluate existing methods across a number of axes, investigating replay ratios, model sizes, and energy functions.

**Strengths:**

- The library is well-motivated, as speeding up RL leads to better experimentation
- The authors implement many environments and energy functions
- The scale, energy function, update-to-data ratio experiments are interesting and useful for future work on GCRL

**Weaknesses:**

The library appears like a "one-and-done" sort of thing that will not be maintained after publication. In RL there is already a large graveyard of abandoned RL projects that no longer run and provide no value to the community. Given this fact, I can only review the current state of the library. In its current state, I think the library needs a bit more work before publication. Please see https://docs.cleanrl.dev for an example of what I think a modern RL library should look like.

- There is no documentation, it is unclear:
    - Which approaches are implemented
    - How to use these approaches
    - How to add new models
    - The structure of the codebase
- There are no unit tests, so the correctness of the code (and the ability to maintain the code as time goes on) is unclear
- The train script is solely for the authors, relying on a pre-existing conda environment
- There are no tutorials beyond a single bash command that runs a parameter sweep
- The library relies on wandb, and does not seem to run without a wandb account
- As far as I understand, the authors only implement 3 algorithms, and I would like to see more than three baselines so that we can do proper comparisons

**Questions:**

- Figure 4 typo: "though DPO policies remain at the goal for a shorter"

---

> ### Author Response · Authors · 2024-11-17
>
> We thank the reviewer for their time and for reviewing our manuscript. It seems like the reviewer's main concern is facilitating the repository's adoption and ensuring that it is maintained. As one step towards addressing this concern, we have improved the repository's usability by adding new documentation, providing several examples, and enhancing the README.md file. It is worth mentioning that, internally, we have seen several new collaborators adopt our codebase for use in their research. This includes researchers from 5 institutions and one company. Helping these new users adopt the code has also improved the benchmark, resulting in 10+ pull requests and 6 new environments. **Together with the discussion below, does this address the reviewer's concerns about the benchmark?** If not, we are happy to revise the paper further and add additional features to the benchmark. We look forward to continuing the discussion!
>
> > Long term plan for maintaining the project
>
> We appreciate the reviewer’s feedback. We recognise that only a few reinforcement learning (RL) packages stand the test of time, but we are committed to making JaxGCRL one of them. Our package complements the excellent CleanRL library, aiming to raise research standards specifically in the goal-conditioned RL (GCRL) field. Currently, there is no single GCRL benchmark that researchers and practitioners can readily use to test new ideas. Training in previous studies is typically slow [1], the number of tasks is often limited [2], and the tasks tend to be too easy [3].
>
> As evidence of the impact of the work, collaborators from both industry and academia are eager to use the codebase for their own research efforts. Since the initial submission, we have added several new features (e.g., new environments, refactoring, etc) to help onboard new users. We hope that this also helps underscore our commitment to long-term project maintenance.
>
> > There is no documentation, it is unclear:
> > - Which approaches are implemented
> > - How to use these approaches
> > - How to add new models
> > - The structure of the codebase
>
> We thank the reviewer for this suggestion. We have added detailed information about the codebase structure, implemented approaches and environments to the README.md file ([see anonymous code](https://anonymous.4open.science/r/JaxGCRL-2316/README.md)). Additionally, we started [MkDocs documentation](https://anonymous.4open.science/r/JaxGCRL-2316/docs/index.md), which we will host on the project website to make it easier to get started with JaxGCRL.
>
> > - There are no unit tests, so the correctness of the code (and the ability to maintain the code as time goes on) is unclear
> > - The train script is solely for the authors, relying on a pre-existing conda environment
> > - There are no tutorials beyond a single bash command that runs a parameter sweep
> > - The library relies on wandb, and does not seem to run without a wandb account
>
> We started implementing these valuable suggestions, starting with different experiment examples and a flag for *optional* logging to wandb.
>
> > - As far as I understand, the authors only implement 3 algorithms, and I would like to see more than three baselines so that we can do proper comparisons
>
> We've added 3 baselines: TD3, TD3+HER and PPO to the benchmark figure ([Figure 3](https://anonymous.4open.science/r/paperJaxGCRL-EFEB/Accelerating_Goal_Conditioned_RL_Algorithms_and_Research_rebuttal.pdf)).
>
>
> [1] https://github.com/google-research/google-research/tree/master/contrastive_rl
>
> [2] https://github.com/dingyiming0427/goalgail
>
> [3] https://github.com/martius-lab/HiTS

---

> > ### Comment · Reviewer_MvLy · 2024-11-20
> >
> > Thank you for adding the baselines, updating the README, and beginning work on the documentation. This is a good start, but I think there is still quite a bit more work to be done before this library is polished enough for public release. As a show of good faith, I am prepared to update my score if the reviewers promise to complete the following by the camera ready deadline:
> >
> > + Host the documentation on `readthedocs` or alternative
> > + Write proper documentation for each loss function in losses.py (link to the original paper, what it is doing, how it works)
> > + Provide docstrings for all user-facing functions
> > + Annotate all configuration variables with their meaning (for example, those in `training.py`)
> > + Make sure the argparser spits out these variables and associated annotations when `--help` is passed to `training.py`
> > + Add a tutorial on implementing a new loss and using a custom model architecture
> > + Add a unit test that checks for breakages in all environments and wrappers that inherit from `brax` (possibly by initializing each and running for a few timesteps)

---

> > > ### Author Response · Authors · 2024-11-20
> > >
> > > We thank the reviewer for this actionable feedback. We have already completed the following improvements:
> > >
> > > > - Write proper documentation for each loss function in losses.py (link to the original paper, what it is doing, how it works).
> > > > - Annotate all configuration variables with their meaning (for example, those in training.py)
> > > > - Make sure the argparser spits out these variables and associated annotations when --help is passed to training.py
> > >
> > > The changes are in the updated [anonymous code](https://anonymous.4open.science/r/JaxGCRL-2316/README.md).
> > >
> > > We are committed to implementing all the remaining proposed changes before the camera-ready deadline.

---

> > > > ### Author Response · Authors · 2024-11-22
> > > >
> > > > Dear Reviewer MvLy
> > > >
> > > > With the rebuttal deadline approaching, we kindly ask if you could strengthen support for the paper, considering the changes we have already implemented and our commitment to addressing all your suggested improvements. Please let us know if you have any further questions, and we will respond promptly.
> > > >
> > > > Kind regards,
> > > >
> > > > Authors

---

### Author Response · Authors · 2024-11-17
**General Response**

We thank the reviewers for their time reviewing our manuscript. The insights provided by the reviewers have allowed us to increase further the quality and readability of our manuscript and JaxGCRL. So far, we have made the changes listed below in response to reviewers' questions and suggestions.

**Improving the JaxGCRL user experience with proper documentation.** We would like to thank reviewer MvLy for their helpful suggestions on the essential components of a modern RL library. In the coming days, we will share updates on the changes to JaxGCRL to enhance its usability, starting with a documentation (MkDocs Material) update today:

- We added a list of implemented environments.
- We added a concise example of using different environments and methods.
- We described the structure of the codebase.
- We added information on how to run experiments without Weights&Biases by setting a run flag
Reviewers can see these new changes on the anonymised repo: https://anonymous.4open.science/r/JaxGCRL-2316/README.md.

**Readability of our manuscript** - following comments from all the reviewers, we have made several changes to our manuscript, which we believe further improve its clarity:

- In Section 2.2 (Accelerating Deep Reinforcement Learning), we clarified JAX's importance for GPU-accelerated environments.
- In Section 5.1 (Experimental Setup), we added the definition of the goal proximity criterion and information about Brax implementation dependency.
- In Section 5.2 (JaxGCRL Benchmark Results), we added new baselines: TD3, TD3+HER and PPO.
- In Appendix (A.6), we have included additional results with scaled-up CRL in a data-rich setting (see below: Extended results on 300M steps for big architecture).
- We also corrected all the typos and missing citations pointed out by reviewers.

These new changes are visible in the updated manuscript: https://anonymous.4open.science/r/paperJaxGCRL-EFEB/Accelerating_Goal_Conditioned_RL_Algorithms_and_Research_rebuttal.pdf. We have highlighted the changes with orange colour, both text and modified figures.

**Extended results on 300M steps for big architecture** - Reviewer 3DLK pointed out that we haven’t evaluated the highest performing architecture (network with 1024 layer width and depth of 4 and layer norm) on every environment while training for 300M steps as in Section 5.5.

We find that these changes can increase the fraction of trials where the agent reaches the goal at least once, but they do not enable the agent to stabilise around the goal (e.g., on 5 tasks, the best agent spends less than 50% of an episode at the goal). When visualizing the rollouts, we observe that the Humanoid agent falls immediately after reaching the goal state, and the Ant Soccer agent struggles to recover when it pushes the ball too far away. The key takeaway is that there is still room for substantial improvement (e.g., can any RL method spend 80% of time steps at goal on Humanoid task), underscoring one dimension on which these tasks are challenging.

Links to figures from this experiment:
- [Time near goal](https://anonymous.4open.science/r/paperJaxGCRL-EFEB/time_near_goal.png)
- [Success rate](https://anonymous.4open.science/r/paperJaxGCRL-EFEB/success_rate.png)

We have also added a new task, Ant Hard Maze. No method reaches the goal in more than 50% of episodes on this new task. We also want to highlight that modifying proposed environments to make them more difficult is straightforward and needs just modifying a single method (`random_target`), which defines the distribution from which goals are sampled.

We believe that these changes increase the quality of JaxGCRL and our manuscript, and again, we are grateful to the reviewers for their suggestions.

---

### Meta-Review · Area_Chair_KGyT · 2024-12-19

**Metareview:**

This paper proposes a new benchmark for self-supervised goal-conditioned RL (GCRL) which is optimized for running on a GPU and can thus yield high-throughout experiments. The authors benchmark a contrastive RL method against other popular RL algorithms and demonstrate strong gains.

The main weaknesses raised during the review process were around differentiation with Brax, and the difficulty of the provided environments. The authors seem to have addressed all the reviewer concerns.

The reviewers are unanimous in recommending acceptance for this work, and after going through the paper and discussion, I agree.
A minor comment is that there is a broken reference in line 33.

**Additional Comments On Reviewer Discussion:**

There was a good discussion between reviewers and authors, and it seems all reviewers were satisfied with the rebuttal and changes provided by the authors.

---

### Decision · Program_Chairs · 2025-01-22

Accept (Spotlight)